# New Mitogenome Features of Philopotamidae (Insecta: Trichoptera) with Two New Species of *Gunungiella*

**DOI:** 10.3390/insects13121101

**Published:** 2022-11-29

**Authors:** Lang Peng, Xinyu Ge, Faxian Shi, Le Wang, Haoming Zang, Changhai Sun, Beixin Wang

**Affiliations:** 1Laboratory of Insect Taxonomy & Aquatic Insects, College of Plant Protection, Nanjing Agricultural University, Nanjing 210095, China; 2Nanjing Institute of Environmental Sciences, Ministry of Ecology and Environment of China, Nanjing 210042, China

**Keywords:** caddisfly, morphology, mitogenome

## Abstract

**Simple Summary:**

The cosmopolitan family of Philopotamidae Stephens 1829 currently includes 26 genera and more than 1500 extant species, and is represented by three subfamilies, Paulianodinae, Chimarrinae and Philopotaminae. In past studies, the genus *Gunungiella* Ulmer 1913 has been subordinate to the subfamily Philopotaminae. The new species in this study provided additional distribution data to the genus and also provided mitogenome data support for evolutionary lineage studies.

**Abstract:**

A total of 14 individuals of Philopotamidae, from China, were examined. Six species in four genera, including two new species of the genus *Gunungiella*, were recognized. Their *COI* barcode sequences were extracted, mitogenomes were sequenced, assembled and analyzed. All of these sequences were used to further reveal the phylogenetic relationships of the family Philopotamidae. In addition, two new species: *Gunungiella wangi* n. sp., *Gunungiella flabellata* n. sp. were described and illustrated.

## 1. Introduction

The Philopotamidae is a caddis family with approximately 1500 species in 26 genera distributed on all continents except Antarctica [1]. Palpi of adults are of considerable length; the maxillary palpi are with a fifth segment forming a flexible flagellum, the second segment is twice the length of the first, and the fourth segment is half the length of the fifth; the ocelli are present; spur formula is 2-4-4 or 1-4-4. Venations are complete, usually with all forks present: I, II, III, IV, and V in the front wings and I, II, III, and V in the hind wings. In the front wings, the thyridial cell is very long, discoidal and the median cells are always closed. In the hind wings, Sc ends at R1, and the discoidal and median cells are open or closed [2].

The genus *Gunungiella* Ulmer, 1913 [3] is a peculiar genus of the family Philopotamidae (Trichoptera). A review of this genus was given by Schmid [2]; subsequently, a number of new species were described by other researchers. The last reported species of *Gunungiella* was in 2011 [4,5]. Currently, the genus *Gunungiella* includes eighty-four species [1], all of which are distributed in the Oriental region. Among them, thirty-four from India by Schmid [2,6]; nine from Malaysia by Huisman, Malicky, Melnitsky and Ivanov [7,8,9]; nine from Thailand by Malicky, Chantaramongkol and Prommi [5,10,11,12,13,14]; eight from Philippe by Ulmer, Banks, Malicky and Mey [15,16,17,18,19,20,21]; eight from Indonesia by Ulmer, Malicky, Melnitsky, Ivanov and Oláh [3,4,9,11,22,23]; eight from Vietnam by Malicky and Oláh [13,24,25]; four from China, before this study, by Schmid, Sun and Gui [2,26,27]; three from Sri Lanka by Bank, Schmid, Chantaramongkol and Malicky [28,29,30]; one from Pakistan by Schmid [31].

The genus *Gunungiella* has been considered to belong to the subfamily Philopotaminae as a highly differentiated lineage, derived from the genus *Wormaldia* by Schmid in 1968. The genitalia of this genus varied greatly among species, so it is impossible to deduce the evolution and phylogeny of the species [2]. In addition, no new *Gunungiella* species have been reported or described in the last decade. It proves difficult to get specimens of the genus, which makes the availability of high-quality DNA-grade specimens even rarer. To date, only eight small molecular fragments from six species have been published in NCBI (https://www.ncbi.nlm.nih.gov, accessed on 11 July 2022), which limits our understanding of the phylogenetic position of *Gunungiella* using multi-marker DNA. Meanwhile, lacking molecular markers hinders the application of environmental DNA metabarcoding technology in water quality monitoring [32,33].

Mitochondria are organelles involved in energy metabolism in eukaryotic cells [34,35]. The mitochondrial genome (mitogenome) has the characteristics of maternal inheritance, low sequence recombination, and fast evolutionary rates [36,37]. Generally, the mitogenome contains 13 protein-coding genes (PCGs), 22 transfer RNA genes (tRNAs), two ribosomal RNA genes (rRNAs), and one control region (CR) [38]. As mitogenomes are considered effective markers for revealing phylogenetic relationships and evolution in many insect groups [39,40,41,42], with the popularization of high-throughput sequencing technology, the mitogenome of Trichopteran has gradually increased. Ge et al. [43] sequenced mitogenomes of four genera of Philopotamidae (*Chimarra* Stephens, 1829; *Wormaldia* McLachlan, 1865; *Kisaura* Ross, 1956; *Dolophilodes* Ulmer, 1909) for the first time and they found a *trnQ*-*trnI*-*trnM* rearrangement pattern in these four genera. However, the mitogenome of the genus *Gunungiella* has not been published, and its genomic structure, nucleotide composition, and substitutional and evolutionary rates are still unknown.

In this study, 14 philopotamids from China were examined, and six species in four genera were identified. Among them, two new species were recognized, described, and illustrated. Meanwhile, we obtained new complete mitogenomes of these six species by high-throughput sequencing, among which those of the genus *Gunungiella* are published for the first time. Then we analyzed the genomic structure, base composition, and evolutionary rates of the new mitogenome, expanding our understanding of the mitogenomes of Philopotamidae. In addition, combined with the published mitochondrial genome of the Philopotamidae, Hydropsychidae and Stenopsychidae, we performed a phylogenomic analysis of the Philopotamidae to explore the phylogenetic position of *Gunungiella*.

## 2. Materials and Methods

A total of 14 philopotamid adults were collected during 2017–2021 with light traps [44] and Malaise traps [45]. Then they were identified to species. Out of six species, two are new to science in Table 1.

### 2.1. Morphological Study

All specimens were stored in 95% ethyl alcohol immediately after collection. The methods used for preparation of specimens followed Xu et al. [44]. Male abdomens used for illustrations were cleared with 10% NaOH solution and heated to 90 °C for 10 min to remove all the non-chitinous tissues. Then the cleaned genitalia were rinsed in distilled water and mounted on a depression slide with lactic acid for examination. Genitalia structure of males were traced with a pencil using a Nikon Eclipse 80i microscope equipped with a camera lucida. Pencil drawings were scanned with an Epson Perfection V30 SE scanner, then placed as templates in Adobe Photoshop v.8.1 software and inked digitally with a Wacom CTL-671 tablet to produce final illustrations. Then each abdomen was stored in a microvial together with the remainder of the specimen in 95% ethanol.

The terminology for wing venation follows Ross [46] and male genitalia follows Schmid [2], as indicated in figures and text.

All types are deposited in the Insect Collection, Nanjing Agricultural University, Nanjing, Jiangsu Province, P.R. China (NJAU).

### 2.2. DNA Extraction, Amplification and Sequencing

All philopotamid samples for DNA extraction were preserved in 100% ethanol and stored at −20 °C. The DNeasy Blood and Tissue kit (QIAGEN) was used to extract the genomic DNA from the legs of each sample according to the manufacturer’s protocol. The DNA of the sample was deposited at the College of Plant Protection, Nanjing Agricultural University, Nanjing, Jiangsu Province, China. The mt*COI* barcoding (685bp) PCR amplification, sequencing, and analysis followed the procedures of Zang et al. [47]. The sequencing libraries were generated for a single sample with an insert size of 350 bp using the genomic DNA at Berry Genomics (Beijing, China). Each sequencing library performed paired-end 150 bp sequencing on the Illumina NovaSeq 6000 platform and then approximately 4–6 Gb clean data were produced for subsequent analysis.

### 2.3. Assembly, Annotation and Nucleotide Composition

NOVOPlasty v3.8.3 [48] was used to de novo assemble the mitogenomes. *COI* barcoding was used as a seed sequence and assembly with k-mer sizes of 39 bp. We predicted tRNAs using a MITOS2 [49] web server with the parameter “invertebrate mitochondrial genetic code”. Geneious 2020.2.1. [50] was used to predict PCGs using Clustal Omega by aligning with the published philopotamid mitogenome. Newly sequenced mitogenomes were submitted to GenBank. SeqKit v0.16.0 [51] was used to calculate the nucleotide composition and base composition of novel mitogenomes. We used DnaSP 6.0 [52] to calculate the rates of the non-synonymous substitution rate (Ka)/synonymous substitution rate (Ks) for each PCG, and the CG View server V 1.0 [53] was used to generate the mitogenome maps. Mitogenome sequences were uploaded to the NCBI database. The accession numbers are shown in Table 2.

### 2.4. Phylogenetic Analyses

To explore the phylogenetic position of the genus *Gunungiella* within Philopotamidae, 26 mitogenomes of Trichoptera were used to reconstruct the phylogenetic tree. We select published 13 hydropsychid and 3 stenopsychid mitogenome as an outgroup, according to the phylogeny of Annulipalpia (Appendix A) [54]. MAFFT version 7.470 [55] was used to align the nucleotide and amino acids of 13 PCGs and two rRNAs genes with the parameter “L-INS-I”. Subsequently, the trimal v1.4.1 [56] was used to remove unreliable homologous regions with “-automated1” method. Finally, FASconCAT-G v1.04 [57] was used to generate five matrixes using the alignments: (1) the PCGFAA (containing amino acid sequences of 13 PCGs); (2) the PCG (containing nucleotide sequences of 13 PCGs). (3) the PCG12 (containing 1st and 2nd codon positions of nucleotide sequences of 13 PCGs); (4) the PCGR (containing nucleotide sequences of 13 PCGs and two rRNA nucleotide sequences); (5) the PCG12R (containing 1st and 2nd codon positions of nucleotide sequences of 13 PCGs and two rRNA nucleotide sequences). We used partitioned maximum likelihood (ML) models and the site-heterogeneous mixture model (CAT+GTR) to infer the phylogenetic relationship of Philopotamidae. For ML analyses, MODELFINDER [58] in IQ-TREE v2.0.7 [59] was used to calculate the best substitution models for each gene. The phylogenetic inference was produced using IQ-TREE with the parameter “-msub mitochondrial -B 1000 [60] --alrt 1000” [61]. To address the heterogeneous effect, the posterior mean site frequency model [62] was used for PCGFAA matrix. For the site-heterogeneous mixture model analyses, we built the phylogenetic tree using Phylobayes-MPI v1.8 [63]. Two independent Markov chain Monte Carlo chains (MCMC) were performed and stopped after the two runs had converged (maxdiff < 0.3). A consensus tree was calculated from the remaining trees combined after the initial 25% trees (burn-in tree) of each chain were discarded [64].

## 3. Results

### 3.1. Taxonomy

#### 3.1.1. *Gunungiella wangi*, Peng, Ge and Sun n. sp. (Figure 1a and Figure 2a–e)

Description: Specimens in alcohol with compound eyes black, mesothorax and legs yellowish brown, abdomen dark brown dorsally, pale yellow ventrally. Fore- and hindwings uniformly brown. Forewings each 4.3 mm long (*n* = 1). Venations typical of genus. Forewings each with fork I, II and V present and sessile; with crossvein between Sc and R_1_, inverted “L”-shaped hyaline stripe across anastomosis, and irregular hyaline area across *m-cu* and base of M_3+4_. Hindwings with fork II and V present, first one petiolate; tip of Sc translucent; with crossvein between Sc and R_1_, and hyaline area across middle part of M; vein A_1_ and A_2_ with base healed, A_1+2_, A_2_ and A_3_ connected to form ring (Figure 1a).

**Figure 1 insects-13-01101-f001:**
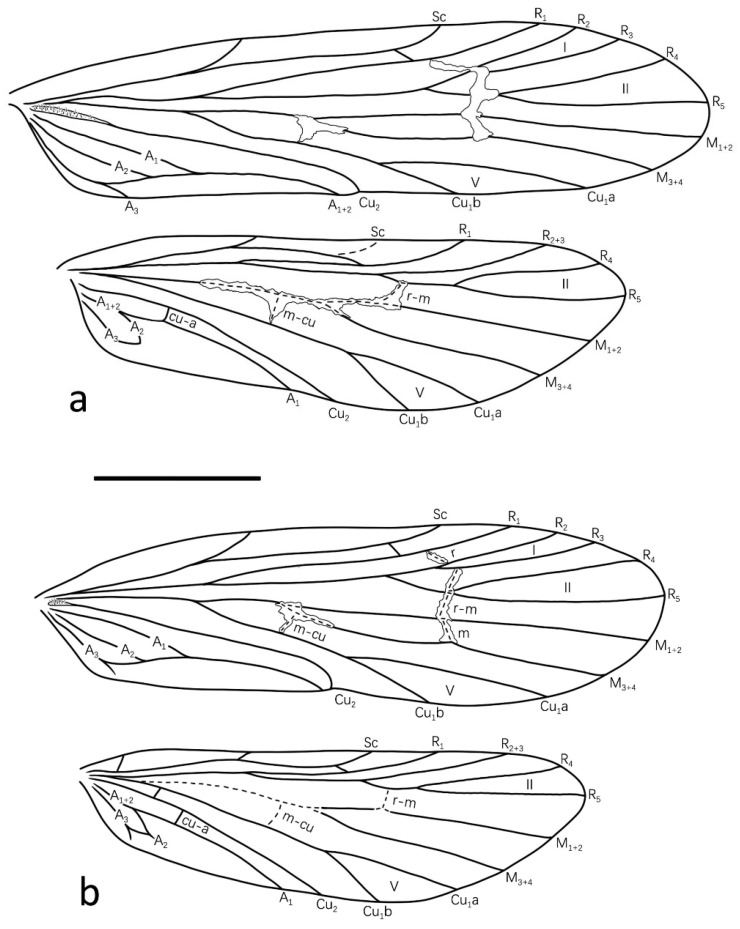
Wing venations of two new species of genus *Gunungiella*: (**a**) *Gunungiella wangi*, Peng, Ge and Sun n. sp.; (**b**) *Gunungiella flabellate*, Peng, Ge and Sun n. sp. Abbreviations: Sc = Subcosta; R = Radius; M = Media; Cu = Cunitus; A = Anal. Scale bar: 1 mm.

Male genitalia: Segment VIII with tergum fused with sternite; trilobate in dorsal view, middle lobe nearly elongate-oval, paired lateral ones each three times the length of the middle one (Figure 2a); nearly square in ventral view (Figure 2b); in lateral view, middle and paired lobes each tapering to apex, lower half of segment VIII nearly rectangular, lateral margins each with upper portions produced into blunt process (Figure 2c). Segment IX deeply invaginated within segment VIII, in dorsal view with upper portion having anterior margin deeply incised in U-shape, and lower portion with posterior margin having shallow incision; in lateral view with anterodorsal portion elongate-triangular, and lower portion rectangular. Segment X narrow and consists of dorsal and ventral branches (Figure 2a); dorsal branch deeply indented at its extremity, producing into two lobes, each lobe with strong spine at apex; ventral branch slightly longer and wider than dorsal one, slightly indented and curved upwards at its extremity (Figure 2a,c). A membranous structure connected segment IX and inferior appendages. First articles constantly fused with each other at base in ventral view (Figure 2b), in lateral view subrectangular, each with dorsal margin arched upwards, articulated with segment IX by anterodorsal (Figure 2c). Second articles smaller than first articles, club-shaped in lateral view and nephroid in ventral view; each with row of strong spines at outer lower base (Figure 2c). Phallus slender and sclerotized, hidden underneath segment X (Figure 2c); consists of basally inflated phallotheca and tubular endotheca with four strong spines in the middle. Diagnosis: The new species somewhat resembles *G. saptadachi* Schmid, 1968 from China in the segment X consisting of two layers, but they still can be distinguished from each other in that: (1) tergum VIII and sternite VIII are fused completely in the new species, but separated in *G. saptadachi*; (2) the paired lateral lobes of segment VIII are long, extending significantly beyond the middle lobe, almost reaching to the distal end of segment X in dorsal view; (3) segment IX slender, with anterior margin extending beyond segment VIII anteriorly in lateral view; and (4) in dorsal view the dorsal branch of segment X with the apex deeply indented; the ventral branch of segment X elongate-triangular, with the apex shallowly incised.

**Figure 2 insects-13-01101-f002:**
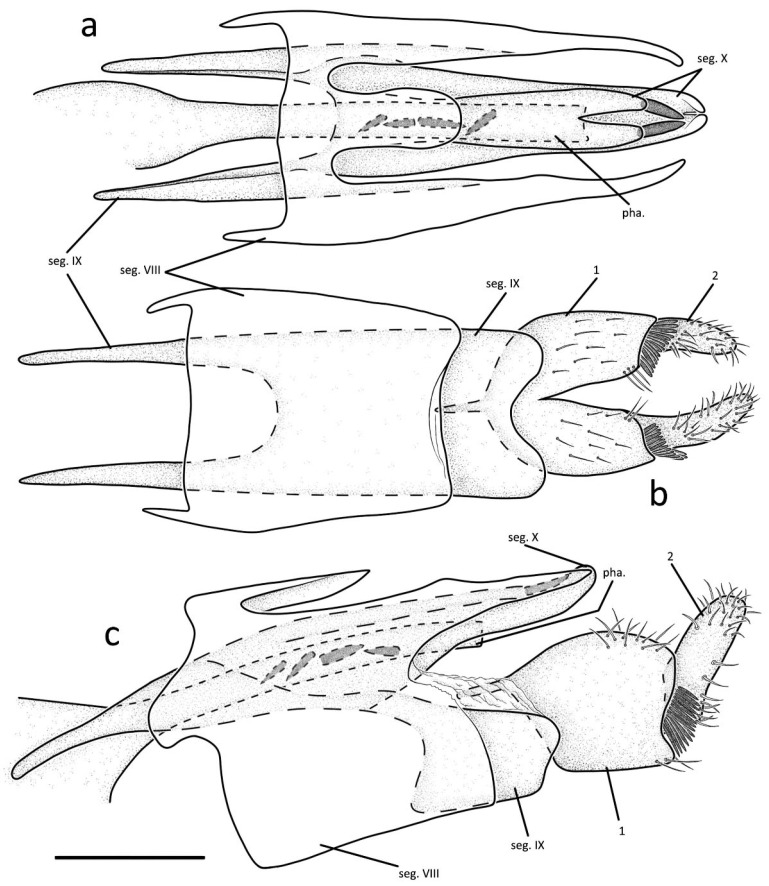
Male genitalia of *Gunungiella wangi*, Peng, Ge and Sun n. sp.: (**a**) dorsal; (**b**) ventral; (**c**) left lateral. Abbreviations: seg. VIII = segment VIII; seg. IX = segment IX; seg. X = segment X; 1 = first article (paired); 2 = second article (paired); pha. = phallus. Scale bar: 0.2 mm.

Holotype: Male, P.R. China, Hu-nan Province, Shao-yang City, Sui-ning County, Guang-yin-xing, 26.4129° N, 110.0898° E, alt. 550 m, 28 May 2021, collected with light traps, coll. L. Peng (NJAU).

Distribution: China (Hu-nan).

Etymology: This new species is named after Dr Xinhua Wang from Nankai University, Tianjin, P. R. China, in honor of his valuable contributions to the field of aquatic biology and in gratitude for all his strong support for Trichoptera research.

#### 3.1.2. *Gunungiella flabellata*, Peng, Ge and Sun n. sp. (Figure 1b and Figure 3a–e)

Description: Specimens in alcohol with compound eyes black; head, mesothorax and legs yellowish brown, abdomen dark brown dorsally, pale yellow ventrally; fore- and hindwings pure brown. Forewing length 3.7–4.0 mm (*n* = 5). Forewings each with fork I, II and V present, first two forks sessile; crossvein between Sc and R_1_, *s*, *r*, *r-m*, *m* and *m-cu* present; with two transparent patches, large one just covers anastomosis, but split at base of branches of R_2_ and R_3_; small one covers *m-cu* and base of M_1+2_ and M_3+4_. Hindwings each with fork II and V present and petiolate; crossvein between Sc and R_1_, *r-m*, *m-cu*, *cu* and *cu-a* present; crossveins *r-m*, *m-cu* and base of M translucent and clear; A_1_ long, each end at hindwing margin; A_2_ and A_3_ short, connected by short crossvein ends, each terminates before wing margin. (Figure 1b).

Male genitalia: Tergum and sternite of segment VIII fused in lateral view, with middle portion of each anterior margin produced into small process; in dorsal view, apex produced into three prominent lobes, middle lobe nearly tapering from base to apex, with apex slightly indented; paired lateral ones less than half length of middle one (Figure 3a); somewhat trapezoid in ventral view (Figure 3b). Segment IX deeply invaginated in segment VIII; in lateral view, upper portion extended anteriad into slender triangular process and lower portion nearly pentagonal; in ventral view, anterior margin deeply incised, posterior margin slightly concave mesally at insertion of first article. Segment X consists of dorsal and ventral branches (Figure 3a); in dorsal view dorsal branch slightly indented apically with two chitinous spines, of same length as middle lobe of segment VIII; paired ventral branches slender, longer than dorsal one, in lateral view with apex curved upwards slightly (Figure 3a,c). First article constantly fused with each other basally in ventral view (Figure 3b), subquadrangularin lateral view, each with dorsoanterior angle produced into slender apodeme and with posterior margin slightly concave at insertion of harpago (Figure 3c). Second article vertical, each with base enlarged with row of strong spines (Figure 3c). Phallus extends anteriad beyond segment IX (Figure 3c); phallotheca sclerotized, with base inflated (Figure 3a,e); endotheca tubular, with three strong spines (Figure 3a,c).

**Figure 3 insects-13-01101-f003:**
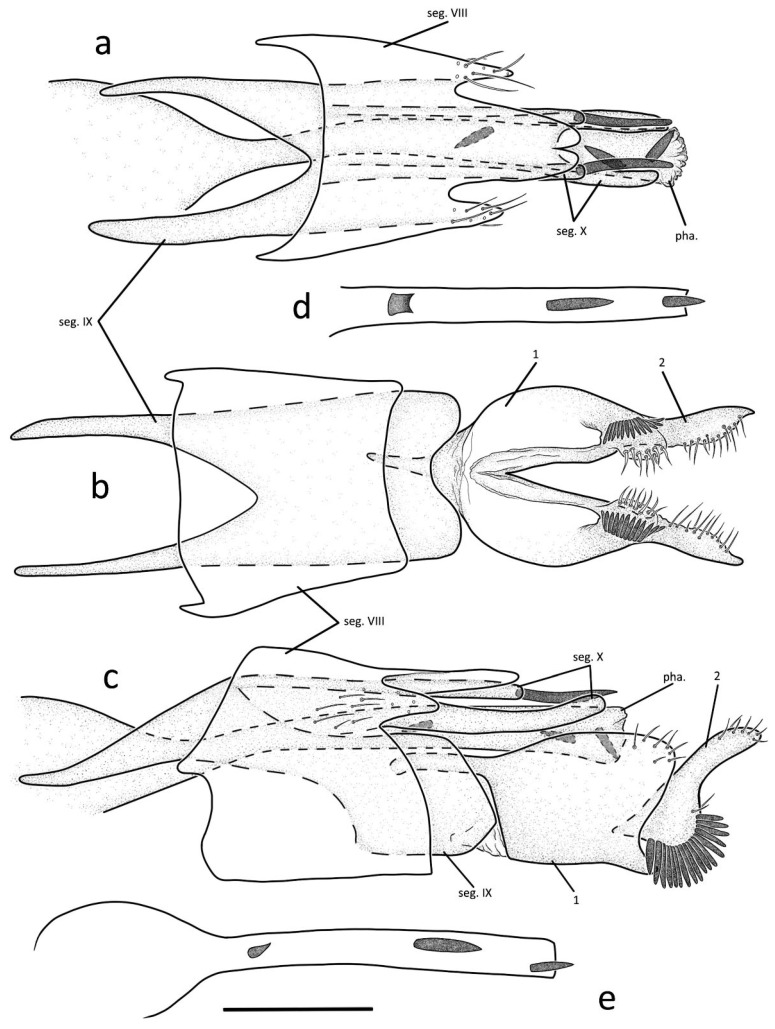
Male genitalia of *Gunungiella flabellata*, Peng, Ge and Sun n. sp., (**a**–**c**) 1 male Holotype specimen, and 7 males from Fu-jian, Da-zhu-lan; (**d**,**e**) 1 male from Fu-jian, Lao-wu farm: (**a**) dorsal; (**b**) ventral; (**c**) left lateral; (**d**) phallus, dorsal; (**e**) phallus, left lateral. Abbreviations: seg. VIII = segment VIII; seg. IX = segment IX; seg. X = segment X; 1 = first article (paired); 2 = second article (paired); pha. = phallus. Scale bar: 0.2 mm.

Diagnosis: The new species resembles *G. saptadachi*, Schmid, 1968 from China in the shapes of the inferior appendage, but it can be distinguished from the latter by: (1) tergum VIII and sternite VIII healing; (2) the middle lobe of segment VIII nearly oval with extremity slightly indented, its length longer than its width, reach the base of the chitinous teeth of segment X dorsal branch; (3) segment IX slender, with anterior margin far exceeding that of segment VIII in lateral view; and (4) ventral branches of segment X paired, slender and curved upwards at the end. This species is also similar to the new species *Gunungiella wangi* n. sp., but it can be distinguished from the latter by paired lateral lobes of segment VIII less than half the length of the middle one, but three times the length of the middle lobe in *Gunungiella wangi* n. sp.

Holotype: Male, P.R. China, Hu-nan Province: Chen-zhou City, Yi-zhang County, Mang-shan Yao Autonomous Township, Mang-shan National Nature Reserve, San-ping waterworks, 24.9818° N, 112.9365° E, alt. 500 m, from 8 to 18 Aug 2020, collected by Malaise trap, coll. X. Lin (NJAU).

Paratypes: 1 male, P.R. China, Fu-jian Province: Wu-yi-shan City, Yang-zhuang Township, Da-an village, Lao-wu farm, 27.8705° N, 117.8636° E, alt. 495 m, 2 July 2021, collected with light trap, coll. C. Sun, X. Ge and L. Peng; 7 males, Wu-yi-shan City, Huang-keng Town, Wu-yi-shan National Park, Da-zhu-lan, 27.6985° N, 117.6521° E, alt. 884 m, 2 to 11 June 2021, collected with Malaise trap, coll. C. Sun, X. Ge and L. Peng (NJAU).

Distribution: China (Hu-nan, Fu-jian).

Etymology: The Latin adjective *flabellatus*, *-a*, *-um* means flabellate referring to the shape of a row of strong spines in the second article in lateral view.

Remarks: Endotheca of seven paratypes from Fu-jian, Da-zhu-lan with three strong spines (Figure 3a,c) as in holotype specimen; endotheca of one paratype from Fu-jian, Lao-wu farm with one tile-like bone and two spines (Figure 3d,e).

### 3.2. New Mitogenome Features of Philopotamidae

Of complete mitogenomes of the six philopotamid species, *Gunungiella flabellata* n.sp. has 36 genes, otherwise, each of the other five species has 37 genes. The mitogenome of *Chimarra sadayu*, *Gunungiella flabellata* n. sp., *Gunungiella acanthoclada*, *Gunungiella wangi* n sp., *Kisaura adamickai* and *Wormaldia spinosa* are 16,602, 15,338, 15,657, 15,333, 15,949 and 15,181 bp in size, respectively (Figure 4). Our results show that the six new mitogenomes with a nucleotide composition similar to the typical insects had a high A + T bias, with A + T content ranging from 73.63% (*Gunungiella flabellata*) to 81.95% (*Kisaura adamickai*) (Appendix A). The highest A + T content was found in the CR, while the lowest was present in PCGs. The A + T content of the genus *Gunungiella* was significantly higher than that of other genera in Philopotamidae (Wilcoxon rank sum test; *p* < 0.05). Most genes (9 PCGs and 14 (13) tRNAs) were encoded on the majority strand (J strand), while the other 14 genes were encoded on the minority strand (N strand). The new mitogenomes of five philopotamids showed positive AT-skew, but that of *Kisaura adamickai* showed negative AT-skew. Meanwhile, the positive GC-skew was also found in all six mitogenomes.

#### 3.2.1. Protein Coding Genes

There was no significant difference in the length of PCGs of the new mitogenomes. *COX3*, *ATP6*, *CYTB*, *ND4* and *ND4L* only had a kind of start codon (ATG); other protein-coding genes had two or three kinds of start codons (Appendix A). All start codons are of the typical ATN mode. Most PCGs had complete termination codons TAA or TAG, while *COX2*, *COX3*, *ND4*, *ND5* and *ND1* of some species have a termination codon TA or T. The A + T content of third codon positions was significantly higher than that of other positions in the PCGs of Philopotamidae. The average ratio of synonymous substitution rate (Ka)/nonsynonymous substitution rate (Ks) (ω) was used to expound the feature of natural selection. The ω values of all PCGs in our study were less than 0.7. Different genes were under different states of purifying pressures, *ATP8* displayed relaxed purifying selection (0.64), while *COX1* was under the strongest purifying selection, with the lowest ω value (0.11) (Figure 5).

#### 3.2.2. Transfer and Ribosomal RNA Genes

Except for *Gunungiella flabellata*, that did not have *trnS2*, the other five new mitogenomes have typical 22 tRNA, ranging in length from 58–73bp. The tRNAs showed high A + T content (83.68–82.78), positive AT-skew and negative GC-skew (Appendix A). We also found a mitochondrial gene rearrangement in our sequenced species. The trnI was translocated to the downstream of *trnQ*, formatting a gene cluster: “*trnQ*-*trnI*-*trnM*”. This rearrangement pattern of tRNA was found in six new mitogenomes. Both rRNAs were encoded on the N-strand of mitogenome and they exhibited similar position and size. The A + T content of *s-rRNA* was significantly higher than that of *l-rRNA* (Wilcoxon rank sum test; *p* < 0.05).

#### 3.2.3. Phylogenetic Analysis

We combined the 20 published mitogenome species (four philopotamid species, 13 hydropsychid species and three stenopsychid species) to explore the phylogenetic relationships within Philopotamidae. The results of AliGROOVE analyses showed the high heterogeneity of the third codon (Appendix A), thus, three datasets were used in this study: (1) the PCGFAA matrix contained 3543; (2) the PCG12 matrix contained 7086; (3) the PCG12R matrix contained 9110 sites. The topologies generated by ML and BI methods demonstrate the monophyly of Philopotamidae. However, two methods generated different phylogenetic relationships within Philopotamidae, ML trees displayed the identical topology (*Gunungiella* + (*Chimarra* + (*Wormaldia* + (*Kisaura* + *Dolophilodes*))))) (Figure 6a); while the BI method showed two different topologies: the one from PCGFAA datasets was inferred as ((*Gunungiella* + *Chimarra*) + (*Wormaldia* + (*Kisaura* + *Dolophilodes*)) and the other from PCG12 and PCG12R datasets was inferred as ((*Chimarra* + (*Wormaldia* + *Gunungiella*)) + *(Kisaura* + *Dolophilodes*)) (Figure 6b,c).

## 4. Discussion

### 4.1. Mitogenome Features

In this study, we obtained six novel mitogenomes of Philopotamidae, belonging to four genera, of which the genus *Gunungiella* was reported for the first time. Significant changes in the length of the mitochondrial genome due to differences in control regions were observed. The nucleotide composition of Philopotamidae had a high A + T bias, which is similar to that of Insecta and Trichoptera [65]. The start codons of protein-coding genes are less diverse than those of other families, and TTG and GTG were not found. The *COX2*, *COX3*, *ND4*, *ND5* and *ND1* of part species appear a termination codon TA or T, which related to post-transcriptional polyadenylation [66,67]. The *COXI* has the lowest ω value, which is consistent with the results observed from other Trichoptera [54]. The *trnQ*-*trnI*-*trnM* rearrangement pattern was found in all newly sequenced mitogenomes, indicating that this may be a synapomorphy rearrangement of the Philopotamid mitogenome. The mitogenome of *Gunungiella flabellata* lacks *trnS2*, which is relatively rare. Generally, tRNAs are the gene category with ‘dispensability’ in the mitogenome; the mitogenomes of some invertebrates do not have the full set of tRNAs [68,69], considering that tRNAs of the cytoplasm were transported to the mitochondria for compensation [70].

### 4.2. Morphology and Mitogenome Implications for the Phylogenetic Position of Gunungiella

By comparing the illustrations of male genitalia and wing venations of all available species of *Gunungiella* with those of other philopotamids, we found that the genus *Gunungiella* shows unique features among its neighbour genera, that is, segment IX is deeply invaginated in segment VIII. In addition, vein M in *Gunungiella* is two-branched both on the fore- and hindwings, which is different from other Philopotamidae genera, in which M is 3- or 4-branched. According to Ge [41], wings are one of the most important organs of insects, and venation modifications are thought to be the successful adaptation to different environmental conditions.

However, the genus *Gunungiella* has been considered to belong to the subfamily Philopotaminae since 1968 [2], and as a highly differentiated lineage derived from the genus *Wormaldia*, based on the morphology. Indeed, the male genitalia shows great similarities between the species of *Gunungiella* and *Wormaldia* in the shapes of inferior appendages and phallus. The inferior appendages in both genera are strongly extended; the first articles are well developed, often connected to the segment IX by a short apodeme; each second articles are relatively smaller, hairy or with spurs and inserts at the apex of the first article, and most of them are usually vertical. Phallus in both genera is simple, club-like, rarely curved; endotheca in some species usually have several strong spines; phallotheca is relatively large, with the base inflated. Under these circumstances, it is unworthy to erect a new subfamily for *Gunungiella*, although the genus *Gunungiella* has some unique features.

To resolve the dilemma, we performed a phylogenetic analysis on mitogenomes. Our results showed that the monophyly of Philopotamidae was recovered in all trees, but the monophyly of the subfamilies was not supported. In the ML tree, *Gunungiella* is inferred as a monophyletic sister group to other philopotamid species, which is similar to the results of the TNT tree by Wahlberg et al. [71]. However, we obtained an unstable topology when using the site-heterogeneous mix model (CAT+GTR); in this situation, the genus *Gunungiella* forms a sister group to *Wormaldia*. Due to incomplete taxon sampling, we cannot fully understand the relationship between each genus. The changes in the phylogenetic position of the genus *Gunungiella* may be a result of long branch attraction [72].

## 5. Conclusions

Both morphological and molecular analyses do not support erecting a new subfamily to accommodate the genus *Gunungiella.* We conservatively treat it as a genus of Philopotaminae. To conclude, the mitogenome data presented in this paper were unable to resolve the deep nodes in the Philopotamidae, so more data are needed to draw firmer conclusions.

## Figures and Tables

**Figure 4 insects-13-01101-f004:**
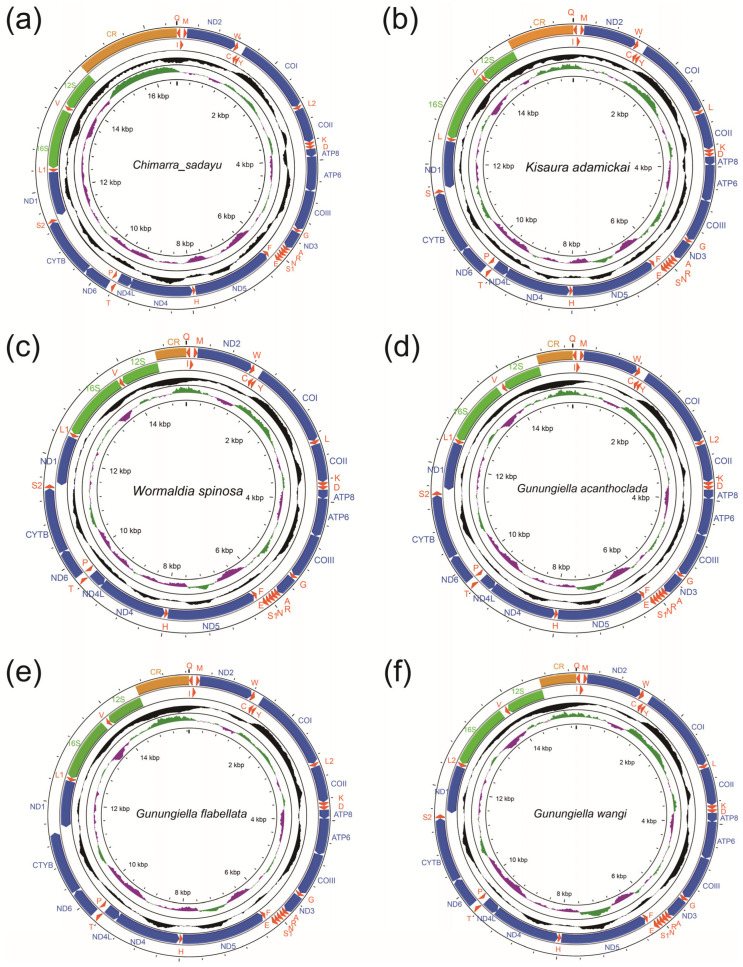
The mitogenome of six species of family Philopotamidae: (**a**) *Chimarra sadayu*; (**b**) *Kisaura adamickai*; (**c**) *Wormaldia spinosa*; (**d**) *Gunungiella acanthoclada*; (**e**) *Gunungiella flabellata* n. sp.; (**f**) *Gunungiella wangi* n. sp. The direction of gene transcription is indicated by the arrows. PCGs are indicated in blue, tRNA in red orange, rRNA genes in green, and CR in yellow.

**Figure 5 insects-13-01101-f005:**
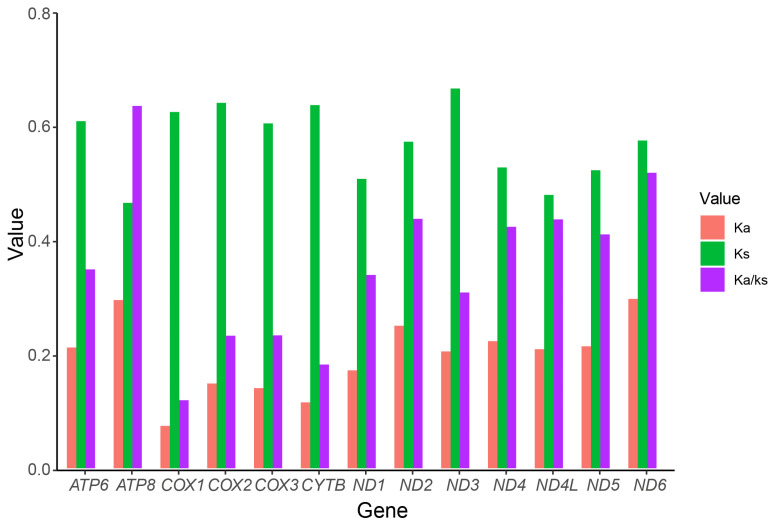
Evolutionary rate of each PCG of the mitogenomes of philopotamid species.

**Figure 6 insects-13-01101-f006:**
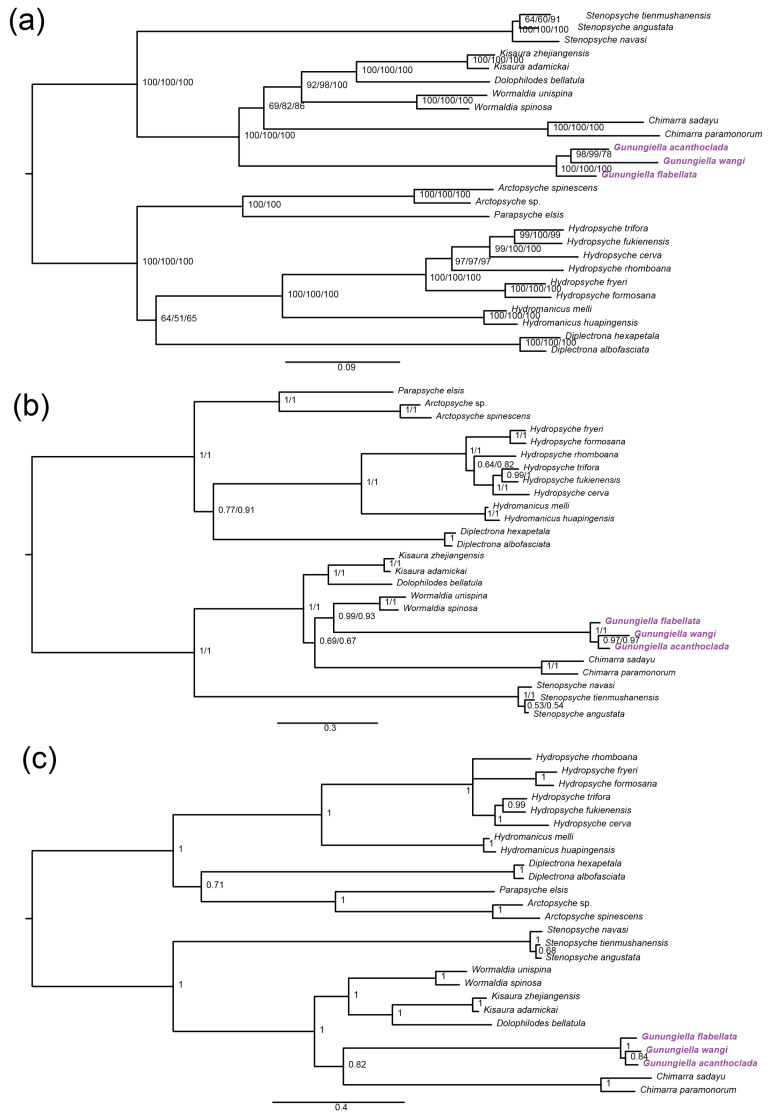
Phylogeny of Philopotamidae as inferred from analyses of the three datasets under the ML model and CAT + GTR mixture model. (**a**) ML tree based on the analysis of three datasets with a partitioning model in IQ-TREE. Node values represent bootstrap probabilities of PCG12, PCG12R and PCGFAA datasets, respectively. (**b**) Phylogeny of Philopotamidae based on PCG12R and PCG12 dataset using CAT+GTR model in phylobayes. Node supports are Bayesian posterior probabilities of PCG12R and PCG12 datasets, respectively. (**c**) Phylogeny of Philopotamidae based on PCGFAA dataset using CAT+GTR model in phylobayes. Node supports are Bayesian posterior probabilities.

**Table 1 insects-13-01101-t001:** Information on the collection of 14 specimens.

Species	Number of Specimens	Collection Site	Trap Type
*Chimarra sadayu* (Malicky, 1993)	1 male	Xian-xi town, Yue-qing city, Zhe-jiang	light trap
*Kisaura adamickai* (Sun and Malicky, 2002)	1 male	Jin-xi town, Long-quan city, Zhe-jiang	light trap
*Wormaldia spinosa* (Ross, 1956)	1 male	Xi-yang town, Yong-an city, Fu-jian	light trap
*Gunungiella acanthoclada* (Sun, 2007)	1 male	Yi-zhang county, Chen-zhou city, Hu-nan	Malaise trap
*Gunungiella wangi* n. sp.	1 male	Sui-ning county, Shao-yang city, Hu-nan	light trap
*Gunungiella flabellata* n. sp.	1 male	Yi-zhang county, Chen-zhou city, Hu-nan	Malaise trap
*Gunungiella flabellata* n. sp.	7 males	Huang-keng town, Wu-yi-shan City, Fu-jian	Malaise trap
*Gunungiella flabellata* n. sp	1 male	Yang-zhuang town, Wu-yi-shan city, Fu-jian	light trap

**Table 2 insects-13-01101-t002:** Specimens used in phylogenetic analyses.

Species	NCBI Accession	Life Stage	Collection Site	Collection Date
*Chimarra sadayu*	OP351513	Adult (male)	Zhe-jiang	27 May 2017
*Kisaura adamickai*	OP351517	Adult (male)	Zhe-jiang	1 July 2021
*Wormaldia spinosa*	OP351518	Adult (male)	Fu-jian	28 June 2021
*Gunungiella acanthoclada*	OP351514	Adult (male)	Hu-nan	8–18 August 2020
*Gunungiella wangi* n. sp.	OP351516	Adult (male)	Hu-nan	28 May 2021
*Gunungiella flabellata* n. sp.	OP351515	Adult (male)	Hu-nan	8–18 August 2020

## Data Availability

The voucher specimens from this research were deposited in the Insect Classification and Aquatic Insect Laboratory, College of Plant Protection, Nanjing Agricultural University, Nanjing, China. The new mitogenomes are deposited in GenBank of NCBI under accession numbers OP351513- OP351518. This published work and the nomenclatural acts it contains have been registered in ZooBank, the online registration system for the ICZN (International Code of Zoological Nomenclature). The LSID (Life Science Identifier) for this publication is: urn:lsid:zoobank.org:pub:E50A14F5-EE27-48F0-A56D-C6403684094C.

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
