# Peer review of "New Mitogenome Features of Philopotamidae (Insecta: Trichoptera) with Two New Species of Gunungiella"

_insects, 2022, doi:10.3390/insects13121101_

Round 1

Reviewer 1 Report

I am wondering about the strange selection of species or specimens for the study. I assume, they were selected to have related genera in the molecular analysis of Gunungiella spp. In this  case, this should be communicated as such.

I am also wondering, why the authors have not included the genus Dolophilodes in their study. Species of this genus are widespread and common, and could have been easily colllected to get fresh material. The genus shows up only  in the phylograms, which means that already available datasets were used.

Author Response

Point 1: I am wondering about the strange selection of species or specimens for the study. I assume, they were selected to have related genera in the molecular analysis of Gunungiella spp. In this case, this should be communicated as such. I am also wondering, why the authors have not included the genus Dolophilodes in their study. Species of this genus are widespread and common, and could have been easily colllected to get fresh material. The genus shows up only in the phylograms, which means that already available datasets were used.

Response 1: Thanks. The facts are as you say, the selection of specimens is for molecular analysis of genus Gunungiella. At the same time, we state in the material that these are all the isoptera specimens that we have collected in recent years for molecular analysis. So far, 18 species of genus Dolophilodes have been recorded in China. Unfortunately, affected by the COVID-19 pandemic, the collection operation has been carried out with great limitations. Thus, only a limited number of available fresh specimens have been collected recent years, and only one that's been successfully sequenced and assembled, it has been published on Ge [41]

Reviewer 2 Report

My suggestion and corrections are indicated in the PDF. The texts is in general well written but there are some odd affirmations that need to be rephrased. The research is interesting and the methods seems to be appropriate. Stamatakis 2016 stated that the use of CAT on datasets with less than 50 taxa is not a good idea because the model can not estimate the per-site rate parameters adequately in this condition. But that was on RaxML, I don’t know if the same will be true for Phylobayes, and several papers have use this model in analyses with few taxa… so is up to the authors to ponder about this. The resulting topologies were inconclusive about the placement of Gunungiella, therefore it seems that the mitogenome was inadequate to solve deep nodes in the family. The taxonomy of the species seems to be ok, the new species illustration are ok.
